# Assessment of the Economic and Social Impact of Shared Parking in Residential Areas

**Xiaofei Ye [1]**, **Xinliu Sui [1]**, **Jin Xie [1]**, **Tao Wang [2,\*]**, **Xingchen Yan [3]** and **Jun Chen [4]**

1   Ningbo Port Trade Cooperation and Development Collaborative Innovation Center, Faculty of Maritime and Transportation, Ningbo University, Ningbo 315211, China; yexiaofei@nbu.edu.cn (X.Y.); suixinliu123@163.com (X.S.); xiejin0428@126.com (J.X.)
2   School of Architecture and Transportation, Guilin University of Electronic Technology, Jinji Road 1#, Guilin 541004, China
3   College of Automobile and Traffic Engineering, Nanjing Forestry University, Nanjing 210037, China; xingchenyan.acad@gmail.com
4   School of Transportation, Southeast University, Dongnandaxue Road 2#, Jiangning Development Zone, Nanjing 211189, China; chenjun@seu.edu.cn
\*   Correspondence: wangtao_seu@163.com; Tel.: +86-152-6785-9815

**Abstract:** Shared parking schemes are not commonly implemented in residential areas due to the uncertainty and conflicts associated with the benefits of such schemes for stakeholders, namely, parking suppliers, parking managers, and the public. To evaluate the economic and social impacts of shared parking in residential areas on its stakeholders, the risk and benefit factors were determined through influential analysis and a questionnaire. A risk–benefit model was established to quantify the risks and benefits for stakeholders. The social return on investment and sensitivity analysis were applied to estimate the economic feasibility of shared parking in residential areas. The methodology combined the use of qualitative, quantitative, and financial information gathered and analyzed to estimate the "value" of shared parking, including its risks, benefits, management pressure, and social benefit. The model was calibrated using the survey data collected from the city of Ningbo in China. The results showed that: (1) The net present value was negative, indicating that the benefits of shared parking were lower than the risks, and thus this scheme would not be economically feasible in residential areas. (2) The cost of purchasing new equipment and rebuilding parking lots had the greatest impact on the benefits of shared parking in residential areas, with a sensitivity coefficient of 4.396, followed by the income from shared parking charges (3.885), and the salary of parking managers (3.619). (3) If the income from parking charges and the salary of parking managers were more than 69,408.5 and 31,091.1 yuan per month, respectively, and the cost of improving parking infrastructure was less than 14,003.2 yuan per month, residential areas could obtain additional benefits due to the acceptance of a shared parking scheme. This study provides theoretical support for the reasonable determination of the costs, risks, and benefits associated with participating in a shared parking scheme in a residential area.

**Keywords:** urban transport; economic feasibility of shared parking; risk–benefit model; residential area; sensitivity analysis

## 1. Introduction

Due to the onset of the economic crisis, the collaborative economy has boomed, resulting in continuous growth of the number of users of these services. The collaborative economy is a model of economic exchange based on three fundamental principles: interaction between producers and

consumers who maintain a continuous dialogue, peer connection due to different technologies (especially digital), and collaboration [1]. Shared parking is a typical problem of the collaborative economy in the field of transportation, and parking problems restrict the sustainable development of cities in China. Shared parking has been applied to address the issue of parking demand and has become a popular research topic in the parking industry and academia [2,3]. In practice, cities such as Ningbo, Shanghai, Beijing, and Guangzhou have attempted to share parking spaces between residential and adjacent commercial areas. However, the implementation of a shared parking scheme in a residential area is challenging. For example, only a few owners are willing to share their parking spaces in the case in Shanghai, and others are unwilling to take the risk of sharing their parking spaces without earning enough benefits. Thus, shared parking schemes have not been widely accepted for several important reasons: (1) Shared parking involves many stakeholders, such as parking suppliers, parking managers, shared parking platforms, government departments, and the public, and different parties have different interests. There are many conflicts and contradictions in risk-taking and benefit distribution, which makes it difficult to promote the mode of shared parking. (2) The risk and benefit categories of shared parking have not been clearly defined or calculated. It is difficult to meet the interests of all stakeholders; taking residential areas as an example, although parking suppliers—who actually own the parking spaces—can make extra income from outside vehicles using their spaces for shared parking, they also suffer risks, including the invasion of privacy and safety concerns. Moreover, although shared parking can effectively solve the problem of parking demand and theoretically create more social benefits, parking managers remain under pressure to maintain the order of external vehicles. Therefore, identifying and quantifying the risks and benefits for stakeholders is an important basis for the promotion of shared parking. Thus, it is of great value to better understand why people accept or reject shared parking. In this study, a risk–benefit model was established to quantify the categories of risks and benefits for stakeholders, i.e., parking suppliers (the owners of parking spaces in residential areas), parking managers (the security personnel of parking lots), and the public. Then, net present value (NPV) and sensitivity analysis methodology were employed to evaluate the economic feasibility of implementing a shared parking scheme in a residential area. The key influencing factors of the benefits of shared parking in residential areas were refined to provide a basis for the theoretical analysis to determine a reasonable cost input, benefit acquisition, and policy formulation by controlling these influencing factors. To the best of our knowledge, this is the first paper to apply the risk–benefit model, NPV, and sensitivity analysis to quantify the economic and social impact of shared parking in residential areas.

## 2. Literature Review

Many scholars have focused on analysis of the utilization characteristics of residential parking spaces, and have explained the feasibility and the methods of implementation from the aspects of shared parking intention [2–4], optimization of resource allocation [5–13], and benefit distribution [14,15].

In terms of shared parking intention, Xie et al. [2] discussed the influence of the benefits and risks on residential parking suppliers' and managers' intentions of implementing a shared parking scheme in a residential area. The results showed that risks and benefits have a significant influence on parking suppliers' intention to implement shared parking, while management pressure has no significance. In addition, benefits, risks, and management pressure have a significant influence on a parking manager's intention of managing a shared parking scheme in a residential area. Meanwhile, Liang et al. [4] showed that perceived control and self-efficacy were the most important factors affecting the use of shared parking by parking demanders.

In terms of the optimization of resource allocation, Chen [7] constructed a dynamic allocation model for the sharing of university parking garages in central urban areas, and obtained a specific time period of opening spaces in universities and residential areas through a test case. Zhen [8] established a two-level programming induction model based on shared parking in residential areas, which was used to measure whether the guidance service can realize the balanced utilization of regional parking resources and whether shared parking is feasible. Yao [9] established the optimal allocation model

of shared space resources in residential areas. Todd [10] studied the differences in the peak hours of parking lots in different types of land. The results showed that the peak hours of parking garages in banks, schools, hospitals, etc., were during week days, while the peak hours of parking garages in entertainment places, cinemas, etc., were at night during the weekend.

In terms of the benefit distribution, Peng et al., [15] analyzed the reasonable benefit distribution expression among shared platforms, community properties, and space owners based on the incomplete information bargaining game model, and used the game model to calculate the reasonable distribution of income. The results showed that the government's support and subsidies for the shared parking strategy can effectively promote the implementation of shared parking schemes. Yang et al. [16] established a relationship model between drivers' intention to accept a parking app, trust in the parking app, their perceived usefulness of the parking app, and their perceived ease of its use. Cai et al., [17] proposed a parking space allocation method by considering the shared parking strategy, parking price, and parking spaces. However, there is still a lack of quantitative analysis on whether it is feasible for residential areas to participate in shared parking schemes from the perspective of economic and social impact. Clarifying the risks and benefits for stakeholders, and assessing the economic and social impact of implementing a shared parking scheme in a residential area, have now become an important basis and prerequisite for promoting shared parking. It is of significant value to analyze the economic feasibility of implementing a shared parking scheme in a residential area to promote such implementation.

## 3. Theoretical Foundation

### 3.1. Net Present Value (NPV)

NPV is a method that determines the sum of money representing the difference between the present value of all inflows and all outflows of cash associated with the project by discounting each at a target rate [18]. The technique uses an external rate of return or discount factor, that is, the expected rate of return on an investment or the discount coefficient assumed based on market investigation and analysis of similar investment performances in the same investment environment. The NPV method represents one of the most established social and economic impact assessment models. Compared to other methods such as qualitative analysis, NPV has the ability to compare the value of different types of benefits and risks, to measure outcomes rather than tracking output, and to provide guidance toward effective and coherent subsidy decisions. Therefore, NPV is capable of evaluating the economic feasibility of implementing a shared parking scheme in a residential area, which is the minimum rate of return that the investor requires to make the investment worthwhile, taking into account the risk involved and all other relevant factors. The annual net cash flow calculation method of parking in a residential area after participating in a shared parking scheme is as follows:

$$E_{NCF,\,n} = (CI - CO)_n (1 - I_{tax}) \tag{1}$$

where $E_{NCF,\,n}$ is the annual net cash value of the $n$th year, $CI$ is the total benefit of residential parking after participating in a shared parking scheme, $CO$ is the total cost of residential parking after participating in a shared parking scheme, and $I_{tax}$ is the income tax on profits. Because shared parking is strongly supported and promoted by the government, profit income tax was not considered, i.e., $I_{tax} = 0$.

The NPV of the mode of shared parking $E_{NPV}$ was calculated according to the annual net cash flow:

$$E_{NPV} = \sum_{t=1}^{n} E_{NCF,t} (1 + i)^t \tag{2}$$

where $i$ is the social discount rate.

$E_{NPV} \geq 0$ indicates that implementing a shared parking scheme is feasible in a residential area. The larger the NPV, the greater the benefit of the investment scheme. By contrast, $E_{NPV} < 0$ means

that implementing a shared parking scheme is unfeasible in a residential area. Then, the relevant influencing factors should be adjusted to ensure that the NPV of the project is greater than zero, and thus make the project operable. In this paper, NPV was used to assess the economic feasibility of participation in a shared parking scheme in a residential area according to the calculation of risks and benefits of shared parking for stakeholders.

### 3.2. Sensitivity Analysis

According to Umeh [19], a sensitivity test is essentially a reappraisal of a project before its commencement using different behavioral criteria or value dimensions for significant variable elements. The first instance is parameterized to arrive at an optimal solution or certain conclusions regarding the viability of the project or scheme being appraised. It tests the robustness of the conclusion of a viability appraisal. It also sheds important light on which variables exert greater individual (or combined) influence on the conclusion of a variability appraisal and the extent to which this influence occurs. Sensitivity analysis is the calculation procedure used for predicting the effect of changes in the key input data on the output results. The magnitude of the effect is expressed by the sensitivity coefficient. In this procedure, input parameters are altered sequentially from their initial values to determine their impact on the outcome of the analysis [20]. This, if necessary, prevents unwanted alterations in the outcome variables. This procedure is often used in investment decision making related to the evaluation of an investment project under uncertain conditions. Generally speaking, the larger the sensitivity coefficient, the greater the impact of the uncertainty on the results of the NPV. The sensitivity coefficient $e$ was calculated according to Equation (3):

$$e = \frac{\Delta E_{NPV} / E_{NPV}}{\Delta F / F} \tag{3}$$

where $\Delta F$ is the variation of an uncertainty factor, and $\Delta E_{NPV}$ is the variation of the evaluation index of the NPV when an uncertainty changes by $\Delta F$.

If $e > 0$, it indicates that uncertainties are positively correlated with the NPV. However, $e < 0$ means that uncertainties are negatively correlated with the NPV. The greater the absolute value of the sensitivity coefficient $e$, the more sensitive the NPV to the uncertainty factor.

This study used the income of the parking charges, the salary of parking managers, and the cost of purchasing new equipment and rebuilding parking lots as the uncertainty factors. The degree of influence of an uncertainty factor on the NPV was analyzed under the conditions of changing the value of an uncertainty factor according to an expected range and keeping the other uncertainty factors unchanged. The values of these uncertainty factors were varied one at a time, while maintaining the other uncertain factors at "base case" values. Each uncertainty factor was varied by ±5%, ±10%, ±15%, and ±20%.

## 4. The Risk–Benefit Model of Shared Parking in Residential Areas

### 4.1. Category and Composition of the Risks and Benefits

This study selected residential parking suppliers, parking managers, and the public as stakeholders. Shared parking increases the profit of parking suppliers and their intention to participate in a shared parking scheme. However, shared parking also brings risks to parking managers and reduces their intention to implement a shared parking scheme. In addition, for the public, the promotion of shared parking effectively solves the social parking problem and brings substantial social benefits. This indicates that the economic feasibility of implementing a shared parking scheme in a residential area is mainly determined by whether the risk and benefit demands of parking suppliers, parking managers, and the public are satisfied. Therefore, according to the collected questionnaire results and the existing literature [2] about shared parking evaluation, the evaluation variables of the risks and benefits for parking suppliers, parking managers, and the public are detailed in Table 1.

**Table 1.** Category and composition of the evaluation variables.

| Stakeholders. | Category | Composition | Description |
|---|---|---|---|
| Parking suppliers | Benefits of shared parking | Economic benefits | • Income can be gained directly through shared parking charges |
| | Risks of shared parking | Cost risks | • The cost of purchasing new equipment and rebuilding parking lots can increase<br>• The salary of parking managers can increase |
| | | Security risks | • The traffic safety of residents cannot be guaranteed<br>• The privacy of residents cannot be guaranteed<br>• Contradictory conflicts can occur between outside vehicles and residents who do not support sharing<br>• Conflicts can occur between parking managers and parking demanders who do not obey said management |
| Parking managers | Benefits of shared parking | Economic benefits | • The salary of parking managers can increase |
| | Risks of shared parking | Management pressure | • The amount of work associated with handling parking conflicts can increase<br>• The amount of work associated with supervising outside vehicles can increase<br>• The amount of work associated with dealing with traffic accidents can increase |
| The public | Benefits of shared parking | Social benefits | • The cruising time for parking demanders can be reduced<br>• Land resources can be saved<br>• The number of employment opportunities can be increased<br>• The amount of illegal parking can be decreased |

### 4.2. Evaluation Variables of Risks and Benefits

4.2.1. Calculating the Risks and Benefits for Parking Suppliers

Economic benefits of shared parking. Parking demanders pay for parking when they use shared spaces in residential areas, and parking suppliers can directly gain economic income. The higher the income, the higher the intention of parking suppliers to share parking spaces. Suppose the income from shared parking charges is $T_J$ (yuan/month, 30 days per month) and represented by Equation (4):

$$T_J = \sum_{j=0}^{k} T_{Jj} \cdot P \times 30 \tag{4}$$

where $k$ is the number of daily parking spaces in residential areas after implementing a shared parking scheme (times), $T_{Jj}$ is the length of parking time of each parking space in the residential area after implementing a shared parking scheme (hours, $j = 0, 1, 2, 3 \cdots k$), and $P$ is the parking charge per hour (yuan/h).

The cost of purchasing new equipment and rebuilding parking lots. There are differences between shared and non-shared parking monitoring devices due to their different functions. New parking monitoring equipment should be added to residential areas, which increases the input cost of implementing a shared parking scheme in a residential area. For parking suppliers, as the cost of purchasing new equipment and rebuilding parking lots increases, the willingness of parking suppliers to share parking spaces decreases. Suppose the cost of purchasing new equipment and rebuilding parking lots is $B_{JC1}$ and represented by Equation (5):

$$B_{JC1} = C_{JBS} + C_{JBR} \tag{5}$$

where $C_{JBS}$ is the cost of purchasing new equipment and $C_{JBR}$ is the cost of the labor.

Management salaries. The salary of parking managers should be appropriately increased due to the increase in management workload, which results in higher labor costs. The results of the questionnaires show that the intention of parking suppliers to share parking spaces is negatively correlated with the cost of management. Thus, suppose the management salary is $S_{JM}$ and represented by Equation (6):

$$S_{JM} = n_{Jm}\left(S_{J0} + \Delta S_J\right) \tag{6}$$

$$\Delta S_J = M_{JT} \times S_{Jh} = O_{Ja} \times n_{Js} \times T_{J1} \times \frac{S_{J0}}{30} \times \frac{1}{8} \tag{7}$$

where $n_{Jm}$ is the number of parking managers after shared parking is implemented in residential areas. $S_{J0}$ is the salary level of parking managers before participating in a shared parking scheme in a residential area (yuan/person/month), $\Delta S_J$ is the increase in the salary level of parking managers after participating in a shared parking scheme in a residential area (yuan/person/month), $M_{JT}$ is the time taken by parking managers to administer external vehicles, $S_{Jh}$ is the salary of a parking manager per hour, $O_{Ja}$ is the turnover of shared parking spaces in residential areas, $n_{Js}$ is the number of parking spaces participating in a shared parking scheme in a residential area, and $T_{J1}$ is the average parking time of a shared parking space.

Loss caused by unsafe traffic for residents. The probability of traffic accidents increases due to the interference of outside vehicles, resulting in the decline of traffic security. For parking suppliers, if traffic security is not guaranteed, willingness to share parking spaces would be lower. Suppose the loss caused by unsafe traffic for residents is $D_{JTr}$ and represented by Equation (8):

$$D_{JTr} = N_{Jm} \times P_{Jtr} \times \overline{B_{Jcr}} \tag{8}$$

where $N_{Jm}$ is the total number of times shared spaces are used per month in residential areas, $P_{Jtr}$ is the probability of traffic accidents after implementing a shared parking scheme in a residential area, and $\overline{B_{Jcr}}$ is the average loss of every traffic accident.

Loss caused by the risk of leaking residents' private information. The parking suppliers' personal information is published to the public, with the risk of personal privacy leakage, which increases the concerns of parking suppliers regarding participation in shared parking schemes. Suppose the loss caused by the risk of leaking residents' private information is $D_{JPs}$ and represented by Equation (9):

$$D_{JPs} = N_{Jm} \times P_{Jpi} \times \overline{C_{JPj}} \tag{9}$$

where $N_{Jm}$ is the total number of times a shared space is used per month in a residential area, $P_{Jpi}$ is the probability of a leak of residents' private information after implementing a shared parking scheme in a residential area, and $\overline{C_{JPj}}$ is the average loss of each accidental leak of residents' private information after implementing a shared parking scheme.

Loss caused by the occurrence of conflicts between outside vehicles and residents. Outside vehicles interfere with the lives of residents who do not support shared parking, which increases the probability of contradictory conflicts between residents and parking demanders, and results in a low intention for parking suppliers to share their parking spaces. Suppose the loss caused by the occurrence of conflicts between outside vehicles and residents is $D_{JPs}$ and represented by Equation (10):

$$D_{JCf} = N_{Jm} \times P_{Jcr} \times \overline{C_{JCf}} \tag{10}$$

where $N_{Jm}$ is the total number of times a shared space is used per month in a residential area, $P_{Jpi}$ is the probability of the occurrence of conflicts between outside vehicles and residents who do not support shared parking, and $\overline{C_{JPj}}$ is the average loss of each occurrence of a conflict between outside vehicles and residents who do not support shared parking.

Loss caused by the occurrence of conflicts between parking managers and parking demanders. Parking demanders who do not obey the management of parking managers cause inconvenience for residents and thus damage the interest of these residents in shared parking schemes. Suppose the loss caused by the occurrence of conflicts between parking managers and parking demanders is $D_{JMl}$ and represented by Equation (11):

$$D_{JMl} = N_{Jm} \times P_{Juo} \times \overline{C_{JMb}} \tag{11}$$

where $N_{Jm}$ is the total number of times a shared space is used per month in a residential area, $P_{Juo}$ is the probability of the occurrence of conflicts between parking managers and parking demanders, and $\overline{C_{JMb}}$ is the average loss of each occurrence of conflicts between parking managers and parking demanders who do not obey the management of these parking managers.

### 4.2.2. Calculating the Risks and Benefits for Parking Managers

Economic benefits of shared parking. Shared parking increases management pressure for parking managers. Suppose the management salary is $S_{JM}$ and represented by Equation (12):

$$S_{JM} = n_{Jm}\left(S_{J0} + \Delta S_J\right) \tag{12}$$

$$\Delta S_J = M_{JT} \times S_{Jh} = O_{Ja} \times n_{Js} \times T_{J1} \times \frac{S_{J0}}{30} \times \frac{1}{8} \tag{13}$$

where $n_{Jm}$ is the number of parking managers after implementing a shared parking scheme in a residential area, $S_{J0}$ is the salary level of parking managers before participating in a shared parking scheme in a residential area (yuan/person/month), $\Delta S_J$ is the increase in the salary level of parking managers after participating in a shared parking scheme in a residential area (yuan/person/month), $M_{JT}$ is the time taken by parking managers to administer outside vehicles, $S_{Jh}$ is the salary of a parking

manager per hour, $O_{Ja}$ is the turnover of shared parking spaces in residential areas, $n_{Js}$ is the number of parking spaces participating in a shared parking scheme in a residential area, and $T_{J1}$ is the average parking time of a shared parking space.

The cost of time for dealing with parking contradictions. Parking managers should appropriately address the parking contradiction between parking demanders and parking suppliers. This tedious work increases the pressure placed on parking managers, which leads to their negative attitude toward shared parking schemes. Suppose the cost of time for dealing with a parking contradiction after implementing a shared parking scheme is $M_{JCt}$ and represented by Equation (14):

$$M_{JCt} = N_{Jm} \times P_{Jco} \times \overline{T_{Jh}} \times W \tag{14}$$

where $N_{Jm}$ is the total number of times a shared space is used per month in a residential area, $P_{Jco}$ is the probability of the occurrence of conflicts between outside vehicles and residents who do not support shared parking, $\overline{T_{Jh}}$ is the average amount of time a parking manager spends dealing with each parking conflict, and $W$ is the cost per unit of time.

The cost of time consumed by the parking managers' supervision of external vehicles. Since the spatial and temporal resources of parking in residential areas are disclosed to the public, the risk of a leak of the private information of residents increases. Therefore, parking managers need to strengthen the supervision and management of outside parked vehicles. Suppose the cost of time consumed by the parking managers' supervision of external vehicles is $M_{JSt}$ and represented by Equation (15):

$$M_{JSt} = O_{Ja} \times n_{Js} \times T_{J1} \times W \times 30 \tag{15}$$

where $O_{Ja}$ is the turnover of shared parking spaces in residential areas, $n_{Js}$ is the number of parking spaces participating in a shared parking scheme in a residential area, $T_{J1}$ is the average parking time of a shared parking space, and $W$ is the cost of time per unit.

The cost of time for dealing with traffic accidents. The higher the number of outside vehicles, the greater the impact on the internal traffic in residential areas, which increases traffic congestion and the number of traffic accidents in residential areas. Parking managers need to provide guidance to parking demanders, avoid traffic congestion and accidents, and ensure smooth and safe traffic in residential areas. Suppose the cost of time for dealing with traffic accidents is $M_{JTs}$ and represented by Equation (16):

$$M_{JTs} = N_{Jm} \times P_{Jtr} \times \overline{T_{Jn}} \times W \tag{16}$$

where $N_{Jm}$ is the total number of times a shared space is used per month in a residential area, $P_{Jtr}$ is the probability of traffic accidents after implementing a shared parking scheme in a residential area, $\overline{T_{Jn}}$ is the average amount of time a parking manager spends dealing with each traffic accident, and $W$ is the cost of time per unit.

### 4.2.3. Calculating the Benefits for the Public

Shared parking in residential areas could open up a large amount of parking supply and improve the overall social benefits and living standards of residents. The social benefits are mainly manifested in the following aspects.

Time value generated by reducing cruise time after implementing a shared parking scheme. Residential garages are shared to provide additional parking, which enables parking demanders to more quickly find a space and reduces the associated parking costs. Suppose the time value generated by reducing cruise time is $B_{CPt}$ and represented by Equation (17):

$$B_{CPt} = \sum_{v=1}^{30} \left( W \times \sum_{f=1}^{N_{C1}} (C_{Cb} - C_{Ca}) \right) \tag{17}$$

$$W = \frac{GDP}{PT} \qquad (18)$$

where $W$ is the cost of time per unit, $N_{C1}$ is the number of parking demanders, $C_{Cb}$ is the time spent finding parking without using a shared parking space, $C_{Ca}$ is the time spent finding parking when using a shared parking space, *GDP* is the gross regional product per year, $P$ is total annual working population of the region, and $T$ is total annual working hours.

The benefits of saving land resources. On the basis of retaining the number of total parking spaces and ensuring the original parking demand of various types of buildings, shared parking contributes to activating existing parking resources. Reducing the number of new parking lots to be built is an effective strategy to save urban land resources. Suppose the benefit of saving land resource is $B_{Cts}$ and represented by Equation (19):

$$B_{Cts} = A_{Csh} \times V_{CA} \qquad (19)$$

where $A_{Csh}$ is the total area of shared parking spaces (m$^2$) and $V_{CA}$ is the value per unit of floor area (yuan/m$^2$).

The social benefits of increasing employment opportunities. Shared parking provides more employment opportunities for society due to the need of technical personnel to construct and maintain shared parking platforms, as well as additional offline management personnel. Suppose the social benefit of increasing employment opportunities is $B_{Cjo}$ and represented by Equation (20):

$$B_{Cjo} = N_{Cjo} \times B_{Caw} \qquad (20)$$

where $N_{Cjo}$ is the number of jobs created by implementing a shared parking scheme and $B_{Caw}$ represents the social benefits of each employment opportunity.

The social benefits brought by the reduction in illegal parking. Residential parking garages are idle during the day, which has obvious dynamic characteristics complementary to the surrounding parking demand. More parking spaces are provided for parking demanders to alleviate the issue of parking demand through shared parking, thus reducing the proportion of illegal parking occupying road resources, relieving traffic congestion, and improving the order of traffic. Suppose the social benefits brought by the reduction in illegal parking in $B_{Cip}$ and represented by Equation (21):

$$B_{Cip} = O_{Ca} \times n_{Cs} \times P_{Cwt} \times P_{wz} \times 30 \qquad (21)$$

where $O_{Ca}$ is the turnover of shared parking spaces in residential areas, $n_{Cs}$ is the number of parking spaces participating in a shared parking scheme in a residential area, $P_{Cwt}$ is the fine for every incident of illegal parking, and $P_{wz}$ is the probability of illegal parking.

## 5. Case Study Analysis of Shared Parking in Residential Areas

### 5.1. Data Collection

As shown in the Figure 1, the questionnaires were disseminated in 40 residential areas in the Haishu, Jiangbei, Yinzhou, and Beilun districts of Ningbo city. A total of 798 valid questionnaires were collected, of which valid responses numbered 164 from managers, including at least two property managers and administrative staff; 298 from suppliers, consisting of 248 parking space owners and 40 property management companies that own parking spaces in residential areas; and the remaining 336 from the public, including 170 dwellings in residential areas and 166 parking demanders surrounding these residential areas. For the parking suppliers, 51.19% of the respondents were male and 48.81% were female. For the parking managers, 87.56% of the respondents were male and 12.44% were female. The number of male managers far exceeded that of female managers. For the public, 54.19% of the respondents were male and 45.81% were female.

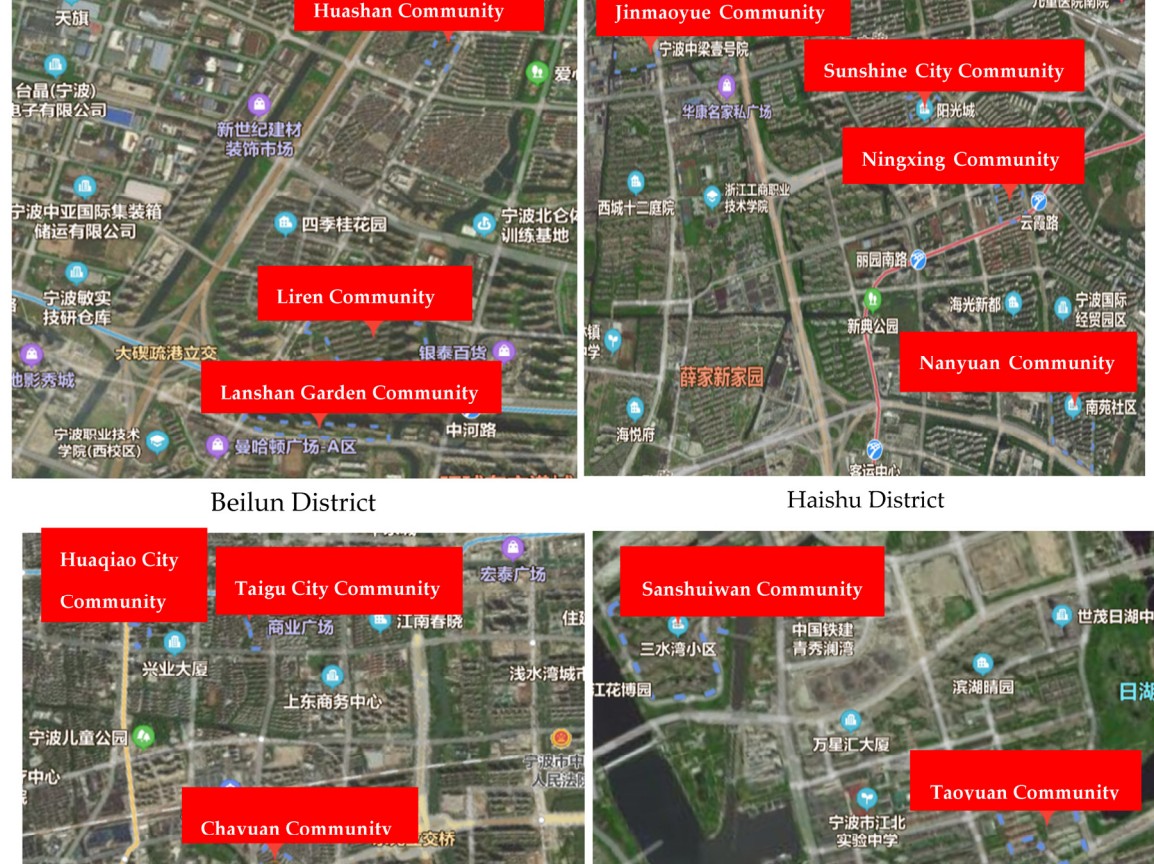

**Figure 1.** Survey areas.

The parking suppliers, parking managers, and the public were asked to indicate what they considered shared parking to be, in addition to the values of the parameters mentioned above, such as the number of parking spaces, acceptable shared parking costs and sharing time, the number of parking managers, the salaries of parking managers, space turnover, and acceptable accident losses. The values of the specific parameters are shown in Table 2.

**Table 2.** List of parameters.

| Parameter | Symbol | Value Range |
|---|---|---|
| Parking charge | $P$ | 4–8 yuan/h |
| Parking time of each parking space | $T_{Jj}$ | 0.5–8 h |
| The cost of purchasing new equipment | $C_{JBS}$ | 135,000–183,000 yuan |
| The cost of labor | $C_{JBR}$ | 100,000–150,000 yuan |
| Salary level of parking managers | $S_{JM}$ | 3000–4500 yuan/month |
| The number of parking managers | $n_{Jm}$ | 4–7 |
| The working hours of parking managers | $M_{JT}$ | 7–10 h/day |
| The number of parking spaces participating in a shared parking scheme | $n_{Js}$ | 100–250 |

**Table 2.** *Cont.*

| Parameter | Symbol | Value Range |
|---|---|---|
| The turnover of shared parking spaces | $O_{Ja}$ | 2–3.5 |
| Probability of the occurrence of traffic accidents after implementing a shared parking scheme | $P_{Jtr}$ | 0.02–0.12% |
| Average loss per traffic accident | $\overline{B_{Jcr}}$ | 4500–6000 yuan |
| Probability of a leak of residents' private information after implementing a shared parking scheme | $P_{Jpi}$ | 0.009–0.055% |
| Average loss per leak of residents' private information | $\overline{C_{JPj}}$ | 5500–10,000 yuan |
| Probability of conflicts between outside vehicles and residents | $P_{Jcr}$ | 0.007–0.029% |
| Average loss per conflicts between outside vehicles and residents | $\overline{C_{JCf}}$ | 6000–11,000 yuan |
| Probability of conflicts between parking managers and parking demanders | $P_{Juo}$ | 0.030–0.050% |
| Average loss per conflicts between parking managers and parking demanders | $\overline{C_{JMb}}$ | 4500–6500 yuan |
| Average time spent by parking managers dealing with each parking conflict | $\overline{T_{Jh}}$ | 2–8 |
| The time spent finding parking without using a shared parking space | $\overline{T_{Jn}}$ | 0.02–0.3 h |
| The time spent finding parking when using a shared parking space | $C_{Cb}$ | 0.01–0.2 h |
| The total area of shared parking spaces | $C_{Ca}$ | 1375–3437.5 m$^2$ |
| Value per unit of floor area | $A_{Csh}$ | 3444.12 yuan/m$^2$ |
| The number of jobs created by implementing a shared parking scheme | $V_{CA}$ | 5–12 |
| The social benefits of each employment opportunity | $N_{Cjo}$ | 1500–3500 yuan |
| The fine for every illegal parking incident | $B_{Caw}$ | 150 yuan |
| Probability of illegal parking | $P_{Cwt}$ | 0.2–5% |
| Cost per unit of time | $P_{wz}$ | 85.184 yuan/h |
| The social discount rate | $W$ | 8% |

According to the effective data from the questionnaires, the average value of each parameter was calculated as the final parameter value in this study. Thus, the final parameter values were as follows:

- The total number of parking spaces in residential areas is 200.
- The shared parking spaces account for 30% of the total parking space.
- The parking charge is 5 yuan/h, and the average parking time in residential areas is 5 h.
- The monthly salary of parking managers in residential areas is 3500 yuan, and the number of parking managers is six. Each month is calculated as 30 days.
- The turnover rate of a parking space before implementing a shared parking scheme is two times per day, while it is three times per day after implementing a shared parking scheme.
- The probability of the occurrence of traffic accidents in residential areas after participating in a shared parking scheme is 0.1%, and the average loss of a traffic accident is 5000 yuan/time.
- The probability of a leak of residents' privacy information after implementing a shared parking scheme in a residential area is 0.05%, and the average loss caused by a privacy leak is 9000 yuan/time.
- The probability of conflicts between outside vehicles and residents who do not support shared parking is 0.02%, and the average loss caused by each conflict is 10,000 yuan/time.
- The probability of conflicts between outside vehicles and residents who do not support shared parking is 0.04%.
- The average loss of each conflict between outside vehicles and residents who do not support shared parking is 6000 yuan/time.

- The probability of conflicts between parking managers and parking demanders is 0.09%, and the cost of time for dealing with parking contradictions is 1022.2 yuan/time.
- The cost of time for the parking managers to supervise external vehicles is 22,999.786 yuan/month.
- The cost of time for dealing with traffic accidents is 5519.949 yuan/month.

Then, by combining this data with the Ningbo Statistical Report—2019, the risks and benefits of shared parking in residential areas was calculated through the risk–benefit model according to Section 3.

### 5.2. Results

The NPV was obtained according to Equation (2).

$$E_{NPV,\ JU} = (-0.4902) * (1 + 8\%)^1 = -0.5295 < 0 \tag{22}$$

The results of the social impact evaluation performed on the mode of shared parking in residential areas show that the discounted impact generated was less than zero. In other words, the mode of shared parking is not economically feasible for residential areas, and the main reason is that the benefits of the current shared parking scheme are far than the risks.

### 5.3. Sensitivity Analysis

To analyze the impact of various uncertainties on the benefits of shared parking in residential areas, a series of one-way sensitivity analyses were undertaken within the range of each parameter in Table 3. The income from parking charges, the salary of parking managers, and the cost of purchasing new equipment and rebuilding parking lots were taken as uncertainty factors. The values of these uncertainty factors were varied one at a time, while maintaining the other uncertainty factors at "base case" values. Each uncertainty factor was fluctuated by ±5%, ±10%, ±15%, and ±20%, and the corresponding change values of the uncertainty factors were obtained. The NPV and the sensitivity coefficient variation statistics are shown in Table 3.

**Table 3.** The statistics of variation of the uncertainty factor, net present value (NPV), and sensitivity coefficient.

| | Uncertainty Factor | | | | | |
|---|---|---|---|---|---|---|
| | The Income from Shared Parking Charges | | The Salary of Parking Managers | | The Cost of Purchasing New Equipment and Rebuilding Parking Lots | |
| Fluctuation | Variation | NPV | Variation | NPV | Variation | NPV |
| −20% | 55,200.0 | −184,142.8 | 24,360.0 | −84,221.2 | 18,126.6 | 53,436.5 |
| −15% | 58,650.0 | −139,430.8 | 25,882.5 | −64,489.6 | 19,259.6 | 38,752.8 |
| −10% | 62,100.0 | −94,718.8 | 27,405.0 | −44,758.0 | 20,392.5 | 24,070.4 |
| −5% | 65,550.0 | −50,006.8 | 28,927.5 | −25,026.4 | 21,525.4 | 9388.0 |
| 0 | 69,000.0 | −5295.00 | 30,450.0 | −5295.0 | 22,658.3 | −5295.00 |
| 5% | 72,450.0 | 39,417.20 | 31,972.5 | 14,436.8 | 23,791.2 | −19,976.8 |
| 10% | 75,900.0 | 84,129.20 | 33,495.0 | 34,168.4 | 24,924.1 | −34,659.1 |
| 15% | 79,350.0 | 128,841.2 | 35,017.5 | 49,907.4 | 26,057.0 | −49,341.5 |
| 20% | 82,800.0 | 173,553.2 | 36,540.0 | 68,177.4 | 27,190.0 | −64,025.2 |
| Sensitivity Coefficient | 3.885 | | 3.619 | | 4.396 | |

As shown in Table 3, the costs of purchasing new equipment and rebuilding parking lots have the largest effect on the benefits of shared parking in residential areas, with a sensitivity coefficient of 4.396, followed by the income from shared parking charges (3.885), and the salary of parking managers (3.619).

As shown in Figure 2, the income from shared parking charges and the salary of parking managers are positively correlated with NPV.

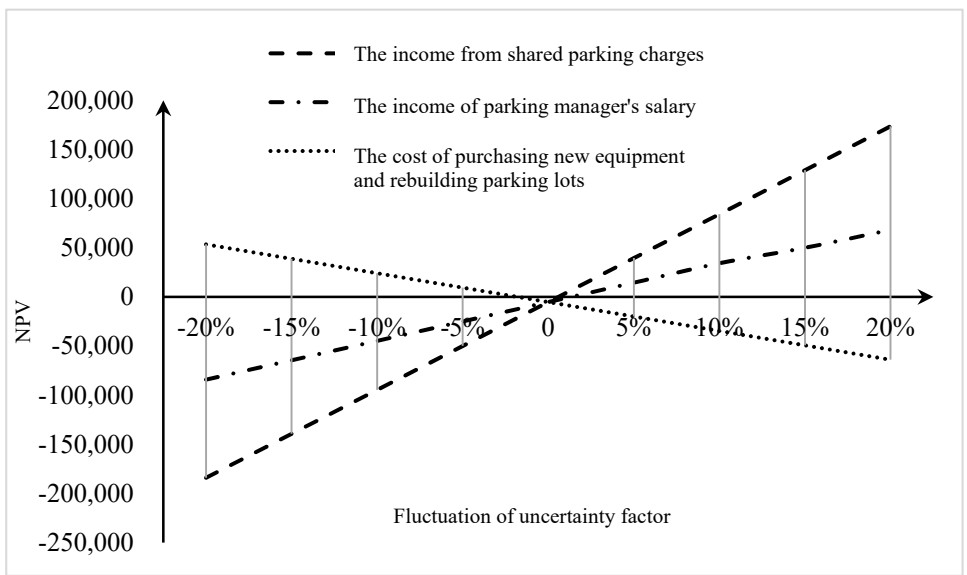

**Figure 2.** Sensitivity analysis of the NPV in residential areas.

The higher the income from shared parking charges and the salary of parking managers, the higher the NPV. However, the cost of purchasing new equipment and rebuilding parking lots is negatively correlated to the NPV. If the cost of equipment transformation is larger, the NPV will be reduced.

Therefore, $E_{NPV, JU} = 0$ means that the shared parking in residential areas results in neither loss nor profit. The critical values of the income from shared parking charges, the salary of parking managers, and the cost of purchasing new equipment and rebuilding parking lots was calculated using Equations (23)–(25):

$$F_{JP1} = \frac{\Delta E_{NPVJP}/E_{NPVJP}}{e_{JP}} * F_{JP0} + F_{JP0} = 69408.5(\text{yuan}/\text{month}) \tag{23}$$

$$F_{JV1} = \frac{\Delta E_{NPVJV}/E_{NPVJV}}{e_{JV}} * F_{JV0} + F_{JV0} = 31091.1(\text{yuan}/\text{month}) \tag{24}$$

$$F_{JC1} = \frac{\Delta E_{NPVJC}/E_{NPVJC}}{e_{JC}} * F_{JC0} + F_{JC0} = 14003.2(\text{yuan}/\text{month}) \tag{25}$$

The results show that when the income from parking charges is more than 69,408.5 yuan/month, the salary of parking managers is more than 31,091.1 yuan/month, and the cost of purchasing new equipment and rebuilding parking lots is less than 14,003.2 yuan/month, shared parking in residential areas can gain additional benefits without bearing additional cost risks. On this premise, it is suggested to implement shared parking schemes in residential areas.

## 6. Discussion

Based on the results of the analysis, it can be observed that the income from parking charges, the salary of parking managers, and the costs of purchasing new equipment and rebuilding parking lots have different effects on the economic feasibility of implementing a shared parking scheme in

residential areas. Therefore, to improve the economic feasibility of shared parking, a balance between the costs and benefits of implementing a shared parking scheme in residential areas should be further proposed from the following perspectives, so as to promote its implementation.

First, an intelligent parking automatic management system should be applied to reduce the management cost, the labor required, and the salary of parking managers. Using automatic photo recognition of an all-in-one vehicle license plate recognition machine, the door to the parking area can be automatically lifted and lowered. The free access of vehicles can reduce the tedious work of management personnel and can save personnel-related costs for parking lots and parking management. Furthermore, although the intelligent parking automatic management system has already been implemented in residential areas, it only provides services for the residents in the community and not the surrounding parking resources. This system cannot connect other parking demanders outside of the community or provide a shared service for all parking demanders. To reduce the information barriers of isolated residential parking resources, the intelligent connected parking management system should be promoted to connect and reallocate shared parking resources.

Second, government subsidies should be increased to balance the benefits and costs of shared parking in parking lots. In cases where the actual revenue of shared parking is lower than the actual cost or the expected revenue, the economic loss value should be calculated according to the difference between the actual cost or the expected revenue and the actual revenue. Government subsidies are a necessary economic stimulus to improve the enthusiasm for and initiative of implementing a shared parking scheme in parking lots. If government subsidies are designed to compensate for the cost of shared parking, shared parking would be widely promoted.

Third, parking charges and the turnover rate of shared parking should be raised synchronously to increase revenue. Aiming at the expected benefit of shared parking revenue of a parking lot, it is suggested that a reasonable dynamic parking pricing model should be adopted to establish an appropriate parking charge standard. In addition, the introduction of a parking guidance system would enable the reasonable allocation of parking resources and social needs, which could effectively improve the turnover rate of parking spaces and bring more benefits to parking lots.

Finally, parking suppliers with high cost performance should be evaluated and selected to reduce the cost input while ensuring the parking quality. Appropriate parking equipment should be selected according to the advantages of this equipment and the actual conditions of specific parking lots. It is also necessary to organize parking space renovation and to build a parking service platform with the highest cost performance.

## 7. Conclusions

This paper aimed to contribute to the economic and social issue of externality evaluation of shared parking in residential areas. Most previous research explained the feasibility and methods of implementation of shared parking from the perspectives of shared parking intention, optimization of resource allocation, and benefit distribution. Unlike the existing literature, this study focused on quantifying the risks and benefits of the stakeholders of shared parking in residential areas, in addition to assessing its economic and social impact.

A risk–benefit model was established to quantify the risks and benefits for stakeholders. The proposed methodology, namely, the NPV, represented a step forward with respect to the risk–benefit analysis, since it directly involved the stakeholders. It analyzed the impact of shared parking on residential areas, and on its stakeholders, in terms of not only economic value produced, but also social dimensions. The model was calibrated by survey data collected from the city of Ningbo in China.

This paper discovered that $E_{NPV,\ JU} < 0$, which indicates that the benefits of the current shared parking scheme are lower than its risks. The social impact evaluation performed on the mode of shared parking in residential areas showed that the discounted impact generated was less than zero, resulting in the infeasibility of shared parking in residential areas. The results showed that shared parking in residential areas obtains additional benefits when the income from parking charges is more than

69,408.5 yuan per month, the salary of parking managers is more than 31,091.1 yuan per month, and the cost of equipment transformation is less than 14,003.2 yuan per month. Regarding the sensitivity analysis, the cost of purchasing new equipment and rebuilding parking lots had the largest effect on the benefits of shared parking in residential areas, with a sensitivity coefficient of 4.396, followed by the income from shared parking charges (3.885), and the salary of parking managers (3.619).

A further step to improve our work may be the inclusion of more parking lots of different land types, which would help to assess the economic and social impacts of shared parking for more types of parking lots, thus enriching our study.

**Author Contributions:** Data curation, X.S., J.X., and X.Y. (Xingchen Yan); funding acquisition, X.Y. (Xiaofei Ye); investigation, J.X.; methodology, J.X.; project administration, X.Y. (Xiaofei Ye); software, X.S.; supervision, T.W. and J.C.; validation, X.Y. (Xingchen Yan); writing—original draft, X.Y. (Xiaofei Ye); writing—review and editing, J.X. and T.W. All authors have read and agreed to the published version of the manuscript.

**Funding:** This research was funded by Natural Science Foundation of Zhejiang Province, China (No. LY20E080011); National Natural Science Foundation of China (grant numbers 71971059, 71701108, and 71861006); National Key Research and Development Program of China—Traffic Modeling, Surveillance and Control with Connected & Automated Vehicles, China (grant number 2017YFE9134700); Natural Science Foundation of Ningbo, Zhejiang Province, China (grant number 2017A610139); Natural Science Foundation of Jiangsu Province, China (grant number BK20180775); and Technology Commission Foundation of Jiangsu Province, China (grant number BK20170932).

**Acknowledgments:** The authors acknowledge the financial support from the relevant institutions. The authors also thank the respondents for providing data and information that were essential for this work.

**Conflicts of Interest:** The authors declare no conflict of interest.

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
