# Peer review of "Assessment of the Economic and Social Impact of Shared Parking in Residential Areas"

_information, doi:10.3390/info11090411_

Round 1

Reviewer 1 Report

Shared parking is a very important issue, due to the car traffic congestion in the cities and the resulting difficulties. The article is important, but needs improvement.

  1. The biggest shortcoming of the manuscript is the lack of information about the conducted surveys. They are a source of data for calculations. The only thing known about the surveys is that "The interview survey was conducted in 40 residential areas in Gulou, Chenghuang temple, Yuehu Shengyuan, and the Yinzhou district of Ningbo city. The suppliers, managers and the public ..." . It is not known how many people took part in them. It is not known how many people were in each group including the suppliers, managers and the public (overall and in each city). It is not known what their socio-economic characteristics were. It is not known when the survey was conducted.
  2. It is not known how the values adopted for the model were calculated. For example, in Table 2, the parking fee parameter has a value range of 4-8 yuan/hour and and for the calculation, the parking fee is 5 yuan per hour.
  3. Are these cities so similar that the results obtained in the survey averaged?
  4. Were all respondents able to answer the questions - e.g. indicate the probability of various events (e.g. probability of illegal parking, probability of the leakage of residents' privacy information after shared parking, etc.).

All ambiguities must be explained in detail.

Other comments:

  1. There should be a graphic illustration showing the location of residential areas of the cities in which the surveys were conducted.
  2. The literature analysis covers only 18 items but there are a rich literature exists on the subject. Literature review should be extended.
  3. The information in the footer refers to the magazine Sustainability ("Sustainability 2020, x, x; doi: FOR PEER REVIEW, www.mdpi.com/journal/sustainability") and the header refers to the journal Information.

Reviewer 2 Report

The authors have evaluated the economic and social impact of shared parking in the residential area.
Risk and benefit factors were determined by sensitivity analysis and a questionnaire survey.
The model was calibrated with data from the survey conducted in the city of Ningbo, China. The results showed that the Net Present Value was negative, indicating that the benefits of carpooling were much lower than the risks and that the current scheme in the residential area would not be economically viable.
The main advantage of this study is that it provides theoretical support for the reasonable determination of the cost, risk and benefit of participating in residential carpooling, and provides a method of quantitative analysis for the promotion of carpooling.
However, the authors have not presented a theoretical background on the meaning of collaborative economics with examples of success stories and studies to support them.
If they did, the reader would be able to better understand the advantages of shared parking in the residential area, since it is not an isolated problem, but rather part of a trend in consumption on a global scale.
I recommend that you consult the following works:
Cheng, M. (2016). Sharing economy: A review and agenda for future research. International Journal of Hospitality Management, 57, 60-70.
Palos-Sanchez, P.R. & Correia, M.B. (2018). The collaborative economy based analysis of demand: study of Airbnb case in Spain and Portugal. Journal of Theoretical and Applied Electronic Commerce Research 13 (3), doi: 0.4067 / S0718-18762018000300105
S. J. Barnes and J. Mattsson, Understanding current and future issues in collaborative consumption: A four-stage Delphi study, Technological Forecasting and Social Change, vol. 104, pp. 200-211, 2016.
C. Anderson, The New Industrial Revolution. New York: Crown Business, 2012.
J. Owyang, C. Tran and C. Silva, The Collaborative Economy. United States: Altimeter, 2013.
The authors present a Net Present Value methodology from Udo, G. O. Model Building in Property Valuation. Enugu: Institute for Development Studies. 2003.
However, they do not explain why it is the most appropriate for this study. What other similar research has used it and what for?
They have conducted a survey, but how did they select the participants? Is it a representative sample? Why this area and city?
In terms of the results of the analysis, the authors find that parking fee revenue, parking revenue, manager's salary, and the cost of purchasing new equipment and rebuilding the parking lot have different effects on the economic viability of participating in shared parking in residential areas. This result is obvious.
Therefore, in order to improve the economic viability of shared parking, a balance between the costs and benefits of shared parking in residential areas should be proposed. How do you propose to do this?
The following perspectives are proposed, in order to promote the implementation of shared parking in the residential area. However, many of them do not bring anything new.
First, an automatic intelligent parking management system should be implemented to reduce the cost of administration, labor and the manager's salary. This system is already in place in many car parks around the world.
Second, government subsidies must be increased to balance the benefits and costs of sharing. The calculation of the government subsidy does not seem to bring anything new from the scientific point of view.
Thirdly, parking costs and car pool turnover rates must be increased synchronously to increase revenues. This perspective is the most interesting, but the authors do not elaborate on the dynamic parking pricing model. But can they indicate how exactly?
Finally, parking equipment providers must be evaluated and selected at a high cost to reduce the cost of entry while ensuring parking quality. This perspective is obvious and does not seem to be new to the reader.
Therefore, the authors should improve the discussion and final conclusions, especially on the aspects explained above. If, in addition, the results showed that the Net Present Value was negative, where does the proposal lie?
I wish the authors good luck in improving this research work.

Round 2

Reviewer 1 Report

Please include the following fragment of the answer to the review in paragraph 5.1. Data Collection  - “The 798 valid questionnaires were collected. For the parking suppliers, 51.19% of respondents were male and 48.81% were female. For the parking managers, 87.56% of respondents were male and 12.44% were female. The number of male managers is far more than female managers. For the public, 54.19% of respondents were male and 45.81% were female”. 

Please add the information on how many people were in each group: suppliers, managers and the public (overall).

Author Response

Response to Reviewer 1

Point 1: Please include the following fragment of the answer to the review in paragraph 5.1. Data Collection  - “The 798 valid questionnaires were collected. For the parking suppliers, 51.19% of respondents were male and 48.81% were female. For the parking managers, 87.56% of respondents were male and 12.44% were female. The number of male managers is far more than female managers. For the public, 54.19% of respondents were male and 45.81% were female”. Please add the information on how many people were in each group: suppliers, managers and the public (overall).

Response: We have added the detail of questionnaire survey into Section Data Collection. Of the 798 valid questionnaires, there were 164 managers including at least two property managers and the administrative staff; there were 298 suppliers consisting of 248 parking berth owners and 40 property management companies which own parking spaces in the residential areas; the others 336 are from the public including 170 dwellings in the residential areas and 166 parkers surrounding the residential areas.

Thank you again for your suggestion.

Reviewer 2 Report

ok

Author Response

thank you again.